# Vemurafenib and Dabrafenib Downregulates RIPK4 Level

**DOI:** 10.3390/cancers15030918

**Published:** 2023-02-01

**Authors:** Ewelina Madej, Anna A. Brożyna, Agnieszka Adamczyk, Norbert Wronski, Agnieszka Harazin-Lechowska, Anna Muzyk, Krzysztof Makuch, Michal Markiewicz, Janusz Rys, Agnieszka Wolnicka-Glubisz

**Affiliations:** 1Department of Biophysics and Cancer Biology, Faculty of Biochemistry, Biophysics and Biotechnology, Jagiellonian University, 7 Gronostajowa Street, 31-387 Krakow, Poland; 2Department of Human Biology, Institute of Biology, Faculty of Biological and Veterinary Sciences, Nicolaus Copernicus University,1 Lwowska Street, 87-100 Toruń, Poland; 3Department of Tumour Pathology, Maria Sklodowska-Curie National Research Institute of Oncology, Krakow Branch, 11 Gancarska Street, 31-115 Krakow, Poland; 4Department of Computational Biophysics and Bioinformatics, Faculty of Biochemistry, Biophysics and Biotechnology, Jagiellonian University, 7 Gronostajowa Street, 31-387 Krakow, Poland

**Keywords:** RIPK4, melanoma, drug targeting, vemurafenib, dabrafenib, targeted therapies

## Abstract

**Simple Summary:**

RIPK4 kinase has oncogenic functions in melanoma and shows high similarity to BRAF protein, making this kinase sensitive to the BRAF inhibitors vemurafenib and dabrafenib. BRAFi downregulates RIPK4 in both BRAF-mutated and wild-type cells. Although downregulation and upregulation of RIPK4 does not affect signal transduction through the BRAF/MEK/ERK pathway in melanoma, it can inhibit cell proliferation and the FAK/AKT pathway.

**Abstract:**

Vemurafenib and dabrafenib are BRAF kinase inhibitors (BRAFi) used for the treatment of patients with melanoma carrying the V600E BRAF mutation. However, melanoma cells develop resistance to both drugs when used as monotherapy. Therefore, mechanisms of drug resistance are investigated, and new molecular targets are sought that could completely inhibit melanoma progression. Since receptor-interacting protein kinase (RIPK4) probably functions as an oncogene in melanoma and its structure is similar to the BRAF protein, we analyzed the impact of vemurafenib and dabrafenib on RIPK4 in melanomas. The in silico study confirmed the high similarity of BRAF kinase domains to the RIPK4 protein at both the sequence and structural levels and suggests that BRAFi could directly bind to RIPK4 even more strongly than to ATP. Furthermore, BRAFi inhibited ERK1/2 activity and lowered RIPK4 protein levels in BRAF-mutated melanoma cells (A375 and WM266.4), while in wild-type BRAF cells (BLM and LoVo), both inhibitors decreased the level of RIPK4 and enhanced ERK1/2 activity. The phosphorylation of phosphatidylethanolamine binding protein 1 (PEBP1)—a suppressor of the BRAF/MEK/ERK pathway—via RIPK4 observed in pancreatic cancer did not occur in melanoma. Neither downregulation nor upregulation of RIPK4 in BRAF- mutated cells affected PEBP1 levels or the BRAF/MEK/ERK pathway. The downregulation of RIPK4 inhibited cell proliferation and the FAK/AKT pathway, and increased BRAFi efficiency in WM266.4 cells. However, the silencing of RIPK4 did not induce apoptosis or necroptosis. Our study suggests that RIPK4 may be an off-target for BRAF inhibitors.

## 1. Introduction

BRAF is a serine/threonine protein kinase that activates the mitogen-activated protein kinase (MAPK) pathway. MAPK/ERK signaling, a key regulator of cell cycle progression, is commonly activated in human tumors through somatic oncogenic mutations in the RAS, RAF, and MEK genes [1]. BRAF mutations induce constitutive activation of the RAF/MEK/ERK pathway in many types of tumors. Constitutively activated ERK stimulates cell proliferation and survival, supporting tumor development and growth as well as the aggressiveness of various malignant neoplasms, for example, melanoma, colon carcinoma, or pancreatic carcinoma [2].

Mutations in the BRAF gene occur in 40–85% of melanoma cases, with the lowest frequency in primary, an increase in metastatic sites and the highest in recurrent melanomas [3,4,5]. Approximately half of advanced melanomas harbor mutations in codon 600 of BRAF (class I), with V600E (a switch from valine to glutamic acid resulting from substitution at the second position of codon 600 of nucleotide 1799 T > A; codon GTG>GAG) being the most common mutation substitution and the most common change (up to 90%) [3]. Other BRAF mutations are predominantly represented by different subtypes of V600 (V600K, V600D, V600R, and V600M), which account for 8–20% of pathogenetic gene sequence variants [1,3]. BRAF class I mutations in melanoma cells are accompanied by the specific clinicopathological characteristics and aggressive behavior of melanomas [6,7]. These mutations increase the kinase activity of BRAF by up to 500 times [8,9].

The identification of BRAF-activating mutations has led to the development of selective inhibitors, including vemurafenib (PLX4032; approved by the Food and Drug Administration (FDA) in 2011) and dabrafenib (GSK2118436, approved by the FDA in 2013), which inhibits the signal transduction of the MAPK cascade. Melanoma cell responsiveness to BRAF kinase inhibitors (BRAFi) may vary depending on the level of expression of the BRAF gene/BRAF mRNA level [10], and approximately 15% of melanomas are characterized by intrinsic resistance mechanisms [11]. Despite an initial response of most melanomas, complete long-term responses are uncommon and patients frequently develop acquired resistance to anti-BRAF monotherapy [12,13]. For that reason, the molecular background of drug resistance is being investigated and several mechanisms have been revealed, including the reactivation of the MAPK pathway, increased signaling of the PI3 kinase/AKT/mTOR pathways, loss of tumor suppressor genes, including phosphatase and tensin homologue (PTEN) and neurofibromin 1 (NF-1), activation of other signaling pathways such as activation of the SRF/MRTF (SRF—serum response factor, MRTF—myocardin-related transcription factor) pathway followed by a oncogene RAC1 mutation, and others [12,14,15,16,17].

RIPK4 (receptor-interacting serine/threonine kinase 4) is a structural homolog of BRAF. Huang et al. showed the structural similarity of BRAF to RIPK2, and given the high similarity of RIPK2 to RIPK4, it is implied that RIPK4 is similar to mutated BRAF [18]. RIPK2 and RIPK4 belong to the RIPK kinases family together with five other members (RIPK1, -3,-5, -6, and-7) [19]. RIPK4 is highly conserved in vertebrate species, with shared sequence identities of 90% between human and murine and bovine orthologs and 62% with zebrafish [18,20,21]. Based on sparse published data, it has been suggested that RIPK4 can function as a tumor suppressor as well as an oncogene [21] since in some cancers, a high expression of RIPK4 is related to the development of tumors and correlates with a worse prognosis [21,22,23,24], while in others, RIPK4 levels decrease with tumor progression [25,26]. The role of RIPK4 in melanoma is limited to our recent studies indicating an oncogenic role for this kinase in melanoma. Involvement of RIPK4 in the NFκappaB transduction pathway affects migration, invasive potential and proliferation of melanoma cells [27]. We showed that the level of RIPK4 is increased in some melanoma cells that may represent the invasive front of the tumor, and that regulates PKC/NFkappaB signaling [27]. RIPK4 activates the RAS/RAF/MEK/ERK pathway in pancreatic cancer by promoting the degradation of phosphatidylethanolamine binding protein 1 (PEBP1) [24]. PEPB1, also known as the RAF kinase inhibitor protein (RKIP), is a tumor suppressor that inhibits the RAF1-MAPK and NFkappaB signaling pathways and the G protein-coupled receptor (GPCR) kinase, resulting in the suppression of tumor progression and metastasis [28,29]. The high expression of PEPB1 in melanocytes and the nevi and low expression in malignant melanoma are well documented [30,31]. RIPK4 is a key element of the Wnt/b-catenin signaling pathways [32] or PKC/NFkappaB [23,27,33] signaling pathways, which are important for melanoma cell biology and may therefore serve as a ‘bypass’ mechanism for drug resistance [34,35].

Although the involvement of RIPK4 in cell differentiation and in various steps and signaling pathways of carcinogenesis have been described so far, no binding partners or coactivators for RIPK4 have been identified. Thus, to investigate the relationship between BRAF and RIPK4 in this study, we first verified the similarity of RIPK4 to wild-type BRAF and mutated BRAF. We also studied the interaction of BRAFi with RIPK4 via molecular docking and the effects on RIPK4 levels in wild-type BRAF (LoVo, BLM) and BRAF-mutated cells (A375, WM266.4). Additionally, we tested the impact of RIPK4 downregulation and upregulation/overexpression on the RIPK4/PEPB1 axis and the RAS/RAF/MEK/ERK and FAK/AKT pathways in melanoma cells carrying V600E and V600D mutations. Finally, we analyzed the correlation between RIPK4 and BRAF mutations in clinical samples. To our knowledge, this is the first report showing that RIPK4 kinase does not directly affect signal transduction through the BRAF/MEK/ERK pathway in melanoma and it may be an off-target for BRAF inhibitors.

## 2. Materials and Methods

### 2.1. General Chemicals and Reagents

The BRAF^V600^ kinase inhibitors: dabrafenib (GSK2118436; Cat. No. S2807) and vemurafenib (PLX4032; Cat. No. S1267) were purchased from Selleck Chemicals (http://www.selleckchem.com, accessed on 12 May 2021). U-0126, a phospho-ERK1/2 inhibitor, (Cat. No. U120), Thiazolyl blue tetrazolium bromide (MTT; Cat. No. M5655), Hoechst 33258 (Cat. No. 94403), protease inhibitor cocktail (Cat. No. P8340), PhosSTOP phosphatase inhibitor cocktail (Cat. No. 4906845001), Polybrene (Cat. No. TR1003), blasticidin (Cat. No. 15205), RIPA buffer (Cat. No. R0278), Bicinchoninic Acid Protein Assay Kit (Cat. No. B9643), penicillin 150 U/mL, and streptomycin 100 µg/mL (Cat. No. P4333) were from Sigma-Aldrich (St. Louis, MO, USA). The 12% Bis-Tris gels—TGX™ FastCast™ Acrylamide Kit (Cat. No. 1610175) and Clarity Western ECL substrate (Cat. No. 1705060) were from Bio-Rad, USA. PVDF membranes (0.2 μm pore size; Cat. No. ISEQ00010) were from Millipore. AnnexinV-FITC early apoptosis detection kit (Cat. No. 6592S) and 7AAD (Cat. No. 00-6993-50) were purchased from Thermo Fisher Scientific (USA). M254 medium (Cat. No. M254500) and human melanocyte growth supplement (HMGS; Cat. No. S005) were from Invitrogen, USA. Bovine serum (BSA; Cat. No. ALB001) was from BioShop, Burlington, ON, Canada, Fetal bovine serum (FBS; Cat. No. 10500064) was from Gibco. MycoAlert PLUS Mycoplasma Detection Kit (Cat. No. LT07-118) and DMEM (Cat. No. 12-604F) were from Lonza, Bend, OR, USA. Methanol, formaldehyde, and DMSO were purchased from POCH (Poland). All chemicals were dissolved in DMSO prior to treatment, with concentrations not exceeding 0.05%.

### 2.2. Clinical Samples

Formalin-fixed, paraffin-embedded samples from 68 patients with diagnosed melanoma were histopathologically evaluated. The samples were collected at the archive of the Department of Tumor Pathology, Maria Sklodowska-Curie National Research Institute of Oncology, Krakow Branch from 2019 to 2021 as a part of diagnostic and treatment planning. Each patient signed a consent to use the redundant tissue after diagnostic procedures for the research study before the surgery was performed. However, tissue samples were anonymized and coded to prevent outsiders from identifying individual patients and their tissue samples. The descriptions of the patients included in this study are presented in Table 1. This investigation was carried out following the rules of the Declaration of Helsinki of 1975 (revised in 2008) and the study was approved by the Bioethics Committee of Jagiellonian University (no. 1072.6120.125.2017, date of first approval 28 September 2017, prolong till 31 December 2025).

### 2.3. Detection of BRAF Mutations from Paraffin-Embedded Tissues

DNA was extracted from formalin-fixed, paraffin-embedded tissues. Before extraction, a pathologist (A.H.-L., J.R.) evaluated the tissue slides to confirm the diagnosis of melanoma, paraffin blocks with the most representative histological material were selected for molecular analysis, and the percentage of tumor cells in the area selected for the testing was established. DNA was extracted from tissue sections using a semiautomatic method using Maxwell^®^ RSC Instrument (Promega, Walldorf, Germany). The Maxwell^®^ RSC DNA FFPE kit (Cat. No. AS1450, Promega) was used to isolate genomic DNA using paramagnetic particles. The purity and concentration of isolated DNA were assessed with a NanoDrop 2000c (Thermo Scientific, Waltham, MA, USA). The real-time polymerase chain reaction (qRT-PCR) was performed on a Cobas z480 apparatus (Roche, Basel, Switzerland). The mutation status in codon 600 of BRAF was confirmed with the BRAF Codon 600 Mutation Analysis Kit II (Cat. No. BRAFX-RT64, EntroGen, Inc., Los Angeles, CA, USA). This assay is designed for the detection of BRAF mutations: V600E/E complex (GTG>GAG/GTG>GAA), V600K (GTG>AAG), V600R (GTG>AGG), V600D (GTG>GAT), V600M (GTG>ATG), and V600G (GTG>GGG) in human genomic DNA from tumor tissues of patients diagnosed with melanoma. The assay was performed, and the data was analyzed in accordance with the manufacturer’s instructions.

### 2.4. Immunohistochemistry for RIPK4 and Section Assessment

RIPK4 in tissue samples was detected using immunohistochemistry as previously described [27]. Sections were evaluated under the BX41 microscope (Olympus Optical Co., Tokyo, Japan), the ColorView III camera (Soft Imaging System, Hanover, Germany), and analySIS 3.2 software (Soft Imaging System, Hanover, Germany). The sections were evaluated semi-quantitatively, as previously described [27]. Since we observed heterogenous, uniform, and granular staining, the assessment was performed separately for each of the staining patterns.

### 2.5. Cell Culture

Human melanoma cell lines A375 carrying the BRAF^V600E^ mutation were obtained from the American Type Culture Collection (CRL_1619, ATCC, Manassas, VA, USA) in 2020. WM115, WM266.4 melanoma cells carrying the BRAF^V600D^ mutation and BLM (wild-type for BRAF) melanoma cells were kindly provided by the Department of Medical Biochemistry of Jagiellonian University Medical College (Kraków, Poland) in 2006 and were authenticated using STR profiling in 2021 STR using Identifiler Plus (ABI) and an ABI 3130xl Genetic Analyser (Applied Biosystems, Waltham, MA, USA). The frequency of the mutant BRAF allele in A375 and WM266.4 cells was determined using an external Sanger sequencing service performed by the certified company Genomed S.A. (Appendix A). All melanoma cells were cultured in RPMI1640 medium supplemented with 10% FBS and antibiotics (penicillin 150 U/mL, streptomycin 100 µg/mL) at 37 °C, in a 5% CO_2_ and 95% humidity environment. Normal adult human melanocytes were purchased from Lonza in 2020 (Cat. No. CC-2586, Lonza, USA) and cultured in M254 medium supplemented with human melanocyte growth supplement HMGS. The LoVo human colorectal cancer cell lines (Cat. No. CCl_229, ATCC, USA) were cultured in DMEM supplemented with 10% FBS and 4.5 g of L-1 glucose. Cells were tested negative for the presence of mycoplasma using the MycoAlert PLUS Mycoplasma Detection Kit using a multi-mode microplate reader CLARIOstar Plus (BMG LABTECH, Ortenberg, Germany).

### 2.6. Transfection with Small Interfering RNA (siRNA)

Melanoma cells were seeded in 35mm cell culture dishes at a density of 3 × 10^5^ for 24 h prior to transfection. Transfection was performed as previously described [27,36]. In this procedure, two types of small interfering RNA (siRNA) were used: RIPK4-specific Silencer Select siRNAs (ID: s28865 and s28863, Thermo Fisher Scientific, USA) and Silencer Select Negative Control No. 2 (Thermo Fisher Scientific, Waltham, MA, USA, Cat. No. 4390846), which served as a negative control. The level of RIPK4 expression in siRNA transfected cells was verified using Western blot or qRT-PCR analysis.

### 2.7. Overexpression of RIPK4

A375 cells were transduced with CRISPRa Lentiviral Blast-dCas9-VPR particles with hEF1a promoters (Cat. No. VCAS11922, Horizon Discovery, Waterbeach, UK) at a multiplicity of infection (MOI) 0.3 using polybrene and then selected with 15 µg/mL blasticidin for days. One day before transfection, stable cells were seeded on a 24-well plate at a density of 2 × 10^4^ cells per well. Cells were transfected with four different synthetic sg RNAs (crRNA: tracrRNA targeting the RIPK4 gene; Cat. No: PQ-005308-01-0002, Horizon Discovery, UK) or with non-targeting control (crRNA: tracrRNA; Cat. No: U-009500-01-05, Horizon Discovery, UK) using DharmaFECT 4 Transfection Reagent (Cat. No. T-2001-02, Horizon Discovery, UK) according to the manufacturer’s protocol. RIPK4 expression levels in A375-dCas9-VPR-cr RNA transfected cells were verified after 72 h of transfection using Western blotting.

### 2.8. Western Blotting

Cells were lysed using RIPA buffer supplemented with Complete Protease Inhibitor Cocktail and PhosphoSTOP Phosphatase Inhibitor Cocktail. Cell extract concentration was measured using the Bicinchoninic Acid Protein Assay Kit following the manufacturer’s protocol. 20 µg of total protein extracts were loaded onto 12% Bis-Tris gels (TGX™ FastCast™ Acrylamide Kit) for protein separation and transferred to PVDF membranes (0.2 μm pore size). After blocking in 4% in TBST buffer for 40 min, the membranes were cut and incubated at 4 °C overnight with the following primary antibodies (Appendix A). Secondary antibodies used for detection were HRP-conjugated (Appendix A). The detection was performed using Clarity Western ECL substrate with the ChemiDoc detector (BioRad, Hercules, CA, USA). Band intensities were quantified using ImageLab 5.2.1 software. The original uncropped and unprocessed scans of all blots are in the raw data file.

### 2.9. Viability and Apoptosis Assay

Cell viability was analyzed using the MTT test as previously described [37]. Thiazolyl blue tetrazolium bromide (MTT) was added at a final concentration of 500 ng/mL for 1 h. After the medium, MTT crystals were dissolved in DMSO: ethanol (1:1). The absorbance was measured with a microplate reader (Tecan Genios, Mannedorf, Switzerland) at 560 nm.

The activity of caspase 3/7 was determined using the luminescent Caspase-Glo-3/7 Assay kit (Cat. No. G8091, Promega, Madison, WI, USA) according to the manufacturer’s protocol using a multi-mode microplate reader Synergy H1 (BioTek, Winooski, VT, USA). Annexin V-FITC/7AAD (7-amino-actinomycin D) staining was performed using FACS Calibur (BD Biosciences, San Jose, CA, USA) as described [37].

### 2.10. Quantitative Real-Time PCR

Total RNA was isolated and qRT-PCR was performed using a thermal cycler qTOWER3 (Analitik Jena, Jena, Germany) as previously described [27]. All TaqMan primers were purchased from Thermo Fisher Scientific/Invitrogen (Appendix A). Relative transcript levels were quantified using the 2^−ΔΔCt^ method, with GAPDH as the reference gene.

### 2.11. Ki67 Analysis Using Immunofluorescence Staining

The cells were washed with PBS solution with Ca^2+^ and Mg^2+^ ions and fixed with 3.7% formaldehyde for 10 min at room temperature. To permeabilize the cell membrane, cells were incubated with 90% methanol for 5 min. The cells were then washed with PBS buffer and incubated with 3% BSA in PBS for 30 min. Subsequently, the cells were incubated for 40 min with mouse anti-Ki67 antibody conjugated to Alexa 555 (BD Biosciences, USA) at 1:10 dilution in blocking solution (3% BSA in PBS) (Appendix A). Additionally, cell nuclei were stained using Hoechst 33,258 solution at a concentration of 2 μg/mL. Images were captured using a 14-bit cooled CCD DFC360FX camera attached to a Leica DMI6000B microscope (Leica Microsystems, Wetzlar, Germany) and analyzed using LAS AF (Leica Application Suite Advanced Fluorescence) software (2.6.0).

### 2.12. Sequence Similarity-Blast

The BRAF and RIPK4 proteins were compared using the BLASTp program provided by the National Centre for Biotechnology Information. The sequences P15056, Q2TB16, and Q9ERK0 from the UniProt database were used for human BRAF and RIPK4, and mouse Ripk4, respectively. The alignment of the BLASTp protein was performed using default parameters—word size set to 3, matrix BLOSUM62, gap initiation cost 11, gap extension 1, and conditional compositional matrix adjustment (https://support.nlm.nih.gov/knowledgebase/article/KA-03391/en-us (accessed on 4 July 2022).

### 2.13. Structural Similarity and Molecular Docking

Crystal structure of murine RIPK4 kinase domains (PDB code: 5WNI), BRAF (PDB code: 5CSW), BRAF^V600E^ (PDB code: 4XV2) and ligands: Vemurafenib (PDB code: 032), Dabrafenib (PDB code: P06), and ATP (PDB code: ATP) was downloaded from the RCSB protein data bank (RCSB PDB). Structural similarity was performed in Pymol (ThePyMOL* (* The PyMOL Molecular Graphics System, Version 2.4.0 Schrodinger, LLC, New York, NY, USA)) using the alignment command with 5 cycles of refinement. RIPK4 and ligand structures were prepared with Python Molecule Viewer (PMV) [38]. AutoDock Vina 1.1.2 [39] was used to dock BRAF inhibitors into the ATP-binding sites of RIPK4. Three total runs with different exhaustiveness parameters were performed for each ligand, and the lowest energy conformation was selected. PyMOL software was used to visualize the obtained results.

### 2.14. Statistical Analysis

Data are presented as mean ± SD, unless otherwise indicated in the figure legends. Statistical tests used are indicated in figure legends. Statistical analyses were performed using GraphPad Prism software (version 5.0, GraphPad Software, La Jolla, CA, USA). Differences were measured using ANOVA or Student’s *t*-test, with *p* < 0.05 and *p* < 0.01, and considered significant and labeled by an asterisk in the figures.

## 3. Results

### 3.1. RIPK4 Shows Sequence and Structural Similarity to the BRAF Protein

Figure 1a shows the result of comparing two sequences of amino acid sequences: BRAF and RIPK4. The compared sequences contain the regions of the kinase domains, which, according to the UniProt database, in the case of BRAF kinase is located at positions 457–717 of amino acids in the peptide sequence, whereas in the RIPK4 protein, this region falls between 22–286 amino acid positions. In BLAST, these regions were extended with two amino acid residues (region 455–717 in the BRAF protein (Query) and 20–286 in the RIPK4 protein (Sbjct), as seen in Figure 1a). Comparative analysis showed 30% identity in amino acid sequence between human RIPK4 (UniProt: Q2TB16) and human BRAF (UniProt: P15056), and 95% sequence identity in the kinase region between human RIPK4 (UniProt: Q2TB16) and mouse Ripk4 (UniProt: Q9ERK0) (Figure 1c) and 90% in full-length proteins (Appendix A) [40]. The crystallized form of human RIPK4 is not present. Therefore, in our study we used the crystallized structure of the mouse Ripk4 protein. We found that despite of 30% sequence identity, the murine Ripk4 protein (PDB: 5WNI) is structurally highly similar to the wild-type human BRAF protein (PDB: 5CSW) and BRAF^V600E^ mutant (PDB: 4XV2), with root mean square deviation (RMSD) values of 2.493 Å and 2.483 Å, respectively (Figure 1b).

### 3.2. The BRAF Mutation Does Not Affect RIPK4 Expression in Metastatic Melanoma

To investigate whether BRAF protein mutations affect the RIPK4 expression level in metastatic melanoma, we analyzed clinical samples. The descriptive data are shown in Table 1. The tissues of 68 melanoma patients were analyzed and *BRAF*^V600E^ mutation was present in 39.7%, the *BRAF*^V600K^ mutation in 7.4%, and negative (wild-type) for *BRAF* in 52.9%. The mean age of the patients at the time of surgery was 67.2 years (range 28–90 years).

As in our previous work, we observed two different types of staining within the cytoplasm, agranular and granular, both with different intensities (Figure 2a,b). We did not observe statistically significant differences in the RIPK4 protein expression in metastatic melanoma between those with or without *BRAF* mutations (Figure 2b).

### 3.3. BRAF Inhibitors Decrease the Level of the RIPK4 Protein in Melanoma Cells

To test the effect of BRAF inhibitors (BRAFi; that is, vemurafenib—PLX4031 and dabrafenib—GSK2118436) on RIPK4, we used A375 and WM266.4 melanoma cells, which carry the BRAF^V600E^ and BRAF^V600D^ mutations, respectively (Appendix A) and express RIPK4 [27], and as a reference to BRAF^WT^ cells: BLM (melanoma) and LoVo (colorectal cancer). PLX4032 (10 µM) and GSK2118436 (10 nM), as expected, reduce cell viability and inhibit ERK1/2 phosphorylation in BRAF-mutated cells, but not in BRAF^WT^ cells (Figure 3a,b). However, BRAFi decrease RIPK4 levels in all cell lines tested, regardless of their BRAF status (Figure 3b). To determine whether the lack of active ERK1/2 affects RIPK4 levels in cells, they were incubated with U0126 (5 µM), a specific inhibitor of ERK1/2 phosphorylation. As shown in Figure 3c, U0126 completely blocked ERK1/2 activity within 6 h, yet RIPK4 levels remained unchanged even after 24 h.

Furthermore, to confirm that BRAFi act on RIPK4 independently of their effect on ERK1/2 activation, we overexpressed RIPK4 in A375 cells and further treated these cells and the respective negative control with drugs. The results show that the inhibitors do not affect RIPK4 levels in cells with RIPK4 overexpression, although they block ERK1/2 activity (Figure 4). This confirms that the decrease in RIPK4 levels was due to interaction with the inhibitor rather than negative feedback mechanisms, which further supports the results obtained on BRAF^WT^ cells. Furthermore, overexpression of RIPK4 does not reduce the BRAFi efficacy, as indicated by a similar decrease in ERK1/2 activation (Figure 4a,b) and the cell viability of cells with overexpression (RIPK4.sg1, RIPK.sg2) compared to controls (neg.sg) measured at 24 h using the MTT assay (Figure 4c).

Next, using Vina’s AutoDock 1.1.2 program, we investigated the possibility of a direct interaction between BRAFi and murine RIPK4, as human RIPK4 is not available. Vemurafenib as well as dabrafenib can interact directly with the kinase domain of the murine RIPK4 protein (Figure 5a). The determined affinity energy for the inhibitors is −9.4 kcal/mol for dabrafenib and −9.1 kcal/mol for vemurafenib. Both are lower than those of the ATP molecule, which is −6.8 kcal/mol. Moreover, the inhibitors analyzed the bind at the ATP binding site, i.e., in the activation pocket of the protein, and could be stabilized with hydrogen bonds. We selected hydrogen-acceptor pairs within the distance of 3Å as potential hydrogen bonds. It appears that in the case of vemurafenib, such a bond can be formed with amino acid residues Asp161, Asn148, and Cys52 of the protein, and in the case of dabrafenib, with Ala147 and Val28 (Figure 5b).

### 3.4. Neither the Decrease in RIPK4 Using siRNA nor Crisp/Cas9-Mediated Overexpression of RIPK4 Affects PEBP1 or BRAF/ERK/MEK Signaling Pathway in Melanoma Cell Lines

The level of PEBP1 protein may be regulated using the RIPK4 kinase [24]. Since PEBP1 acts as a tumor suppressor in melanoma [30], we investigated a correlation between the levels of RIPK4 and PEBP1 protein levels in melanoma. The RIPK4 and PEBP1 protein in melanoma cell lines were compared to the levels in normal melanocytes. The comparative analysis showed an inverse relationship. Melanocytes in which RIPK4 kinase expression is not observed have the highest levels of PEBP1 protein expression among the cell lines analyzed. PEBP1 protein levels decreased by 61% (*p* < 0.05) in the WM115 cell line derived from the primary lesion, while a threefold increase in RIPK4 expression is observed in the same cells. A significant decrease (about 75% (*p* < 0.05) in PEBP1 expression was observed in the WM266.4 and A375 cell lines, showing the highest levels of RIPK4 (Figure 6a). Therefore, we ask whether the downregulation or overexpression of RIPK4 affected PEBP1 levels in the WM266.4 and A375 cells. The downregulation of RIPK4 using siRNA had no effect on phospho-PEBP1 or on PEBP1 degradation in both cell lines. Furthermore, siRIPK4 did not alter the BRAF/MEK/ERK pathway (Figure 6b,c). Next, we examine whether the overexpression of RIPK4 would reduce the level of phosphorylated PEPB1. For this purpose, we chose A375 cells, characterized by lower levels of RIPK4 than WM266.4, but a similar level of PEPB1. We increased the RIPK4 levels in A375 cells using the CRISPR/Cas9 system and two guide RNAs. However, the upregulation of RIPK4 had no effect on PEBP1 phosphorylation or PEBP1 degradation compared to the negative Cas9 control and did not alter the BRAF/MEK/ERK pathway (Figure 6b,c).

### 3.5. Silencing RIPK4 Altered the CDK4/CDK6/RB and FAK/AKT Pathways in WM266.4, but Not in A375

We recently found that silencing RIPK4 disrupts the WM266.4 cell cycle [27]. To assess the mechanisms of the RIPK4-mediated regulation of cell proliferation in melanoma, we examined whether downregulation of RIPK4 affects the levels of important proteins involved in cell cycle regulation, that is, the cyclin-dependent kinases and retinoblastoma protein (RB1). In WM266.4 cells, RIPK4 silencing caused a 50% decrease in transcript levels for CDK2 and CDK6, accompanied by a 1.9-fold increase for CDK14, 1.6-fold (for RIPK4.si1; *p* < 0.05) and 2.5-fold (for RIPK4.si2; *p* < 0.001) increase in growth suppressor CDK2AP1 (known as p12(DOC-1) (Figure 7a). Despite the downregulation of RIPK4, no apparent changes were demonstrated in the A375 cell proliferation rate; however, statistically significant changes were observed in the levels of mRNAs encoding cyclin-dependent kinases that regulate the cell cycle. Silencing RIPK4 led to a 29% (*p* < 0.001) decrease in the level of CDK2 mRNA encoding CDK2, 47% (*p* < 0.001) decrease in CDK6 mRNA, and 21% increase in CDK14 transcript compared to control cells. Unlike WM266.4 cells, the mRNA level for CDK2AP1 in A375 remained unchanged (Figure 7a). Western blot analysis confirms that the silencing of RIPK4 in WM266.4 cells downregulates CDK2 and the phosphorylated and basic form of the RB1 protein 48–96 h after transfection (Figure 7b). On the contrary, no changes in CDK2 protein levels or the phosphorylated and basic form of RB1 were observed in A375 cells when compared to the control, regardless of changes in CKD2 mRNA levels. Ki67 staining showed that RIPK4 in WM266.4 reduces the number of Ki67+ cells by 40% 72 h after transfection (*p* < 0.001), whereas no change is observed in A375 cells (Figure 7d). These data are consistent with the results obtained using qRT-PCR and Western blot analysis.

Moreover, we tested whether the downregulation of RIPK4 expression affects the level of phosphorylated AKT and FAK proteins, which are also known to be able to control cell proliferation. Figure 7c illustrates that the downregulation of RIPK4 decreases the activity of AKT and FAK levels in WM266.4 while it has no effect on A375 melanoma cells.

### 3.6. RIPK4 Downregulation Sensitized WM266.4 but Not A375 Melanoma Cells to BRAFi Treatment

Vemurafenib or dabrafenib acts on ERK1/2, which is not affected by RIPK4. However, both BRAFi and RIPK4 silencing affect the proliferation of WM266.4 cells. Therefore, we tested whether the downregulation of RIPK4 enhances the action of BRAFi. Indeed, we found that the silencing of RIPK4 sensitized WM266.4 cells to BRAFi, which was not observed in A375 cells. This is confirmed by an enhanced decrease in RB and CDK2 levels and WM266.4 cell viability (Figure 8a,b) upon treatment with vemurafenib or dabrafenib of si.RIPK4 cells compared to the negative control.

### 3.7. RIPK4 Silencing Using siRNA Has No Impact on Apoptosis or Necroptosis in WM266.4 Cells

After showing that the silencing of RIPK4 inhibits the proliferation of WM266.4 cells, we next examined whether this effect is due to an induction of apoptosis or necroptosis in cells. Interestingly, although the downregulation of RIPK4 leads to a 30% increase in mRNA levels for RIPK1, the mRNA levels for RIPK3 are below the detection threshold in both the RIPK4.si1- and neg.si-transfected cells, indicating that RIPK3 is not expressed in WM266.4 cells (Figure 9a), suggesting that these cells are resistant to necroptosis. We did not observe differences in the levels of the transcripts encoding BCL-2 and BAD, regulatory proteins, in cells 48 h after transfection with RIPK4.si1 versus neg.si (Figure 9a). However, an examination of the MCL-1 transcript level revealed a significant decrease in this transcript (approximately 60%) compared to the control. These observations were accompanied by a small and non-statistically significant increase in caspase activity 3/7 (approximately 34%) (Figure 8b) and the percentage of apoptotic cells (5.7% vs. 4.4% in scrambled control) as determined via the Annexin V/7AAD double staining of cells (Figure 9c, the raw data file) 72 h after transfection.

## 4. Discussion

Vemurafenib (PLX4032) and dabrafenib (GSK2118436), small-molecule mutant BRAF kinase inhibitors, are used to treat melanoma patients diagnosed with the BRAF^V600^ mutation. However, melanoma cells develop resistance to both drugs when used as monotherapy, resulting in disease relapse. Therefore, to prevent the development of cell resistance, combination therapy and the mechanisms leading to resistance are actively investigated.

Our attention was drawn to the kinase RIPK4; it has structural similarity to the BRAF protein [18] and is highly expressed in some primary and metastatic melanoma specimens and cell lines [27]. Huang et al. were the first to obtain the crystal structure of the kinase domain of the mouse Ripk4 protein and showed that it is very similar to the human BRAF protein at the structural level [18]. The kinase domain shows high homology within the RIP protein family and between species [40]. The crystallized form of human RIPK4 is not present. Therefore, in our study, we used the crystallized structure of the mouse RIPK4 protein.

First, we confirmed that the RIPK4 structure and sequence of RIPK4 is similar to both the BRAF protein and its mutant form and then verified that the BRAF^V600^ E or K mutation does not affect the expression of the RIPK4 protein in melanoma specimens. Second, we showed that vemurafenib and dabrafenib led to the downregulation of the RIPK4 protein in A375 and WM266.4 melanoma cells that carry BRAF^V600E/D^ mutations. Furthermore, BRAFi reduced RIPK4 levels 48 h after treatment, in BRAF^WT^ cells such as LoVo and BLM. This effect appears to be specific to BRAF inhibitors, as further confirmed by U126, a specific inhibitor of ERK1/2 kinase phosphorylation, which did not affect RIPK4 levels. Moreover, both BRAFs did not affect RIPK4 levels, but blocked ERK1/2 activity in A375 cells with an introduced RIPK4 overexpression. Additionally, BRAFi increased the level of phosphorylated ERK1/2 in BRAF^WT^ cells, compared to cells with the BRAF^V600^ mutation. Tham et al. reported an increase in ERK1/2 activity with vemurafenib in HaCaT cells [41]. The observed decrease in RIPK4 levels may result from the impact of BRAFi on the p63 protein, since RIPK4 is a direct transcriptional target of the p63 protein [42]. An upregulated p63 level was found in MAPKi-resistant melanoma cells [43]. Additionally, by using molecular docking analysis we observed that both BRAFi could directly interact with the kinase domain of the RIPK4 protein. Vemurafenib and dabrafenib were docked at the binding site of the ATP molecule, which, due to the competitive nature of these inhibitors, confirmed the validity of the analysis. Furthermore, the predicted amino acid residues determined for the RIPK4 protein, which can form hydrogen bonds with BRAF^V600^ inhibitor molecules, are crucial for the activity of the catalytic domain of this kinase. The vemurafenib molecule can probably interact with the Asp161 residue, whose mutation leads to a complete loss of catalytic domain activity of both RIPK4 kinase [18,44] and RIPK3 kinase, indicating a high evolutionary conservation of this residue [19]. On the contrary, the dabrafenib molecule can form a hydrogen bond with the Ala147 residue. This residue is involved in ATP binding as well as binding to other inhibitors, such as staurosporine, lestauritinib, TG-100-115, and tozaserib [18]. J-X Li using the radioactive assay showed that dabrafenib selectively inhibits RIP3 activity over RIP1, RIP2, and RIP5 [45], while RIP4 was not studied. Undoubtedly, affinity testing and the mutagenesis of predicted amino acid residues in RIPK4 involved in BRAFi binding require further study to substantiate our in silico observations of BRAFi binding to RIPK4.

Next, we studied the effects of RIPK4 downregulation on PEBP1 and signal transduction through the BRAF/MEK/ERK pathway. PEBP1 is an important MAPK kinase pathway signal transduction control protein that serves as a negative regulator of the RAS/RAF/MEK/ERK pathway signaling cascade. PEBP1 levels decrease dramatically with the progression of melanoma [30,31]. The results obtained in the present study confirm the significantly lower expression level of the PEBP1 protein in melanoma cells compared to melanocytes. Although we showed that the expression of PEBP1 and RIPK4 is inversely correlated in melanocytes and melanoma, suggesting that RIPK4 blocks the degradation of the PEBP1 protein in metastatic melanoma, the downregulation of RIPK4 in melanoma cells had no effect on phosphorylation or upregulation of the PEBP1 protein. Furthermore, the downregulation and overexpression of RIPK4 did not alter the activity of the BRAF, MEK, or ERK1/2 proteins in any of the melanoma cell lines. It confirms that RIPK4 does not affect the PEBP1/RIPK4 axis or the BRAF/MEK/ERK pathway in melanoma. These results are in contrast to studies in pancreatic cancer cells, suggesting that the RIPK4 kinase is involved in the regulation of PEBP1 protein levels and consequently affects signaling through the BRAF/MEK/ERK pathway [24]. However, our results are consistent with skin cancer studies in which RIPK4 acts as a tumor suppressor but does not affect signaling through this pathway [46].

The analyses performed in this study confirm previous reports on the involvement of RIPK4 in the regulation of the cell cycle of WM266.4 melanoma cells [27]. The CDK4/CDK6 pathway is frequently dysregulated in melanoma, leading to excessive proliferation. The cell cycle-regulating kinases CDK4/CDK6 in complex with cyclin D1 and D2 or D3 can phosphorylate RB, thus causing the release of E2F transcription factors and the G1-S cell cycle transition [47]. Here, we demonstrate that the RIPK4 kinase influences the proliferation rate of melanoma cells by participating in the regulation of cyclin-dependent kinase (CDK) expression. Our results showed that when the level of RIPK4 is decreased, the expression of CDK2 and CDK6 kinases is inhibited in the WM266.4 cell. These changes are accompanied by a decrease in RB1 protein phosphorylation and a significant increase in the transcript level for the negative regulator of the CDK2 kinase, that is, CDK2AP1, consequently leading to a decrease in the proliferation rate of these cells. The addition of vemurafenib or dabrafenib to cells silenced for RIPK4 levels of RIPK4 increases the efficacy of BRAFi in WM 266.4 but not A375 cells as assessed using the MTT assay, CDK2, or RB levels. The lack of significant differences in the proliferation rate of A375 cells after the silencing of RIPK4 may be due to compensatory mechanisms that keep proliferation unchanged despite decreased levels of transcripts for CDK2 and CDK6. It appears that such a factor may be the CDK14 kinase, whose levels increase in the WM266.4 and A375 lines after down-regulation of RIPK4 expression. CDK14 kinase has been implicated as an oncogene in the literature. This kinase promotes the transition from the G to M phase of the cell cycle by phosphorylating LDL receptor related protein 6 (LRP6) and further activating the Wnt pathway [48]. Recently, the CDK14 kinase expression has been shown to be under the control of the AKT pathway, which controls cell proliferation, among other pathways [49]. Indeed, we observed a decrease in AKT and FAK activity in WM.266-4 cells when the level of RIPK4 decreased. In some cancers, FAK activates AKT-mTOR signaling to promote growth and progression [50]. Thus, we suggest that decreasing RIPK4 could potentially restrain melanoma growth as well by inhibiting the AKT-mTOR pathway via FAK. It should be noted that WM266.4 cells are characterized by a loss of PTEN expression (phosphatase and tensin homolog), an inhibitory protein of the ATK pathway, while A375 cells, despite the high proliferation rate, show a normal expression of PTEN [51], thus indicating alternative pathways regulating the proliferation of these cells. Therefore, it can be speculated that despite an increase in the CDK14 transcript levels in both analyzed lines, only in A375 can CDK14 translate into a compensatory pathway that affects the proliferation process due to different signaling pathways that regulate this process. However, this hypothesis should be verified in the future. Cyclin-dependent kinases are important regulators of the cell cycle, with specific classes activated at well-defined phases of the cycle. The main kinase that regulates the transition from G1 to S phase is CDK2, which in complex with cyclin E and CDK4/6-cyclin D together phosphorylates the RB1 protein, a proliferation suppressor. The phosphorylation of the RB1 protein leads to the release of factor E2F, which initiates the transcription of genes necessary for cell cycle progression [52,53,54]. Furthermore, recent studies have shown that the CDK2AP1 protein, through the negative regulation of the CDK2 kinase, acts as a tumor suppressor for breast cancer [55]. Cyclin-dependent kinases are interesting targets for anticancer therapy due to their functions and excess activity in tumors; therefore, further research on cyclin-dependent kinases inhibitors is of growing interest [52,53,54].

The results obtained in the present study indicate that the observed reduction in the proliferation rate of melanoma cells under the downregulation of RIPK4 kinase is not related to the targeting of these cells into the necroptosis or apoptosis. Although the RIPK4 kinase can slightly affect the RIPK1 expression level in WM266.4 cells, the lack of RIPK3 expression renders these cells resistant to necroptosis. The lack of expression of the RIPK3 kinase in A375 cells, as well as in other melanoma cell lines, was demonstrated in a study by Geserick et al. The authors of this study indicate that the RIPK3 expression is lost with melanoma progression, protecting tumor cells from necroptosis despite the preserved expression of the RIPK1 kinase [52]. Over the past decade, kinases belonging to the RIP protein family, RIPK1 and RIPK3 in particular, can participate in the regulation of apoptosis and necroptosis [20,56,57,58]. RIPK1 plays a key role in pro-inflammatory signaling associated with tumor necrosis factor receptor-1 (TNFR1; TNF receptor-associated factor) and Toll-Like receptor 3/4 (TLR3/4; Toll-Like Receptors), and participates in the signaling cascade associated with the process of apoptosis and necroptosis [59]. The presence of a death domain at the carboxyl terminus of the RIPK1 kinase may facilitate its interaction with other proteins containing the death domain by promoting the formation of death receptor-related signaling complexes, while the homotypic RIP interaction motif (RHIM), shared by RIPK1 and RIPK3, allows its interaction with other cell death adaptors [59,60]. Unlike RIPK1, the RIPK4 kinase does not appear to directly participate in the apoptosis-related signaling cascade, possibly due to the lack of a death domain in its structure [61].

In fact, no differences were shown between cells, and negative siRNA-transfected cells were shown in caspase 3/7 activities, Annexin-V staining, and levels of transcripts encoding apoptosis regulating proteins (BCL-2 and BAD). On the other hand, a significant decrease in MCL1 under the influence of RIP4 silencing was observed. The lack of involvement of the RIPK4 kinase in processes leading to cell death are consistent with the results obtained from studies on keratinocytes [62] and tongue squamous cell carcinoma [63]. 

However, the silencing of RIPK4 in tongue squamous cell carcinoma cells sensitized them to the actions of cisplatin [63]. Since MCL-1 acts as a master regulator of apoptosis in various human malignancies, including melanoma [64,65], it is not excluded that a decrease in MCL-1 in WM266.4 cells with RIPK4 silencing may affect their susceptibility to BRAFi.

## 5. Conclusions

In summary, RIPK4 kinase does not directly affect signal transduction through the BRAF/MEK/ERK pathway in melanoma (Figure 10). Instead, it may be an off-target for BRAF inhibitors. However, the treatment of tumors in which this kinase plays an oncogenic role, such as colon and ovarian cancer, requires further research especially related to potential side effects of such treatment.

## Figures and Tables

**Figure 1 cancers-15-00918-f001:**
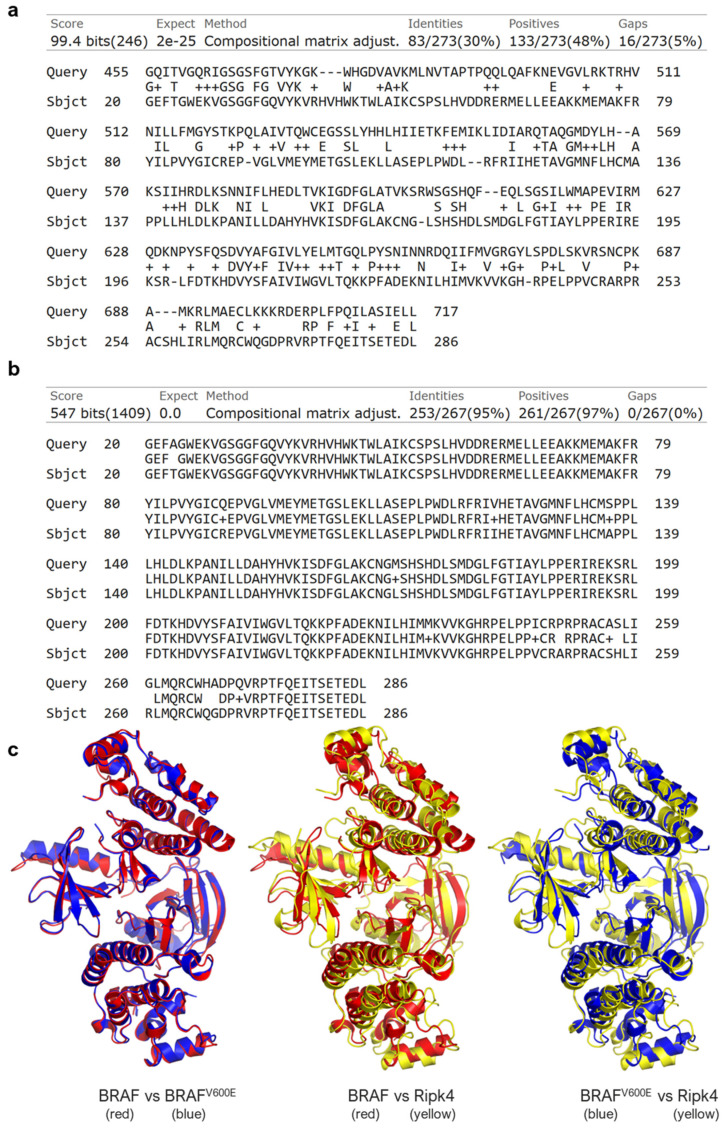
RIPK4 shows the sequence and structural similarity to the BRAF protein. (**a**,**b**) Comparison of amino acid sequences of human RIPK4 (Sbjct) with (**a**) BRAF (Query) or (**b**) mouse Ripk4 (Query) using BLAST. (**c**) Structural alignment of homodimers of human BRAF (red), human BRAF^V600E^ (blue), and mouse RIPK4 (yellow) taken using PyMOL. The root mean square deviation (RMSD) value is 0.334 Å, 2.493 Å, and 2.483 Å for the comparison of BRAF vs. BRAF^V600E^, BRAF vs. Ripk4, and BRAF^V600E^ vs. Ripk4, respectively. his is a figure. Schemes follow the same formatting.

**Figure 2 cancers-15-00918-f002:**
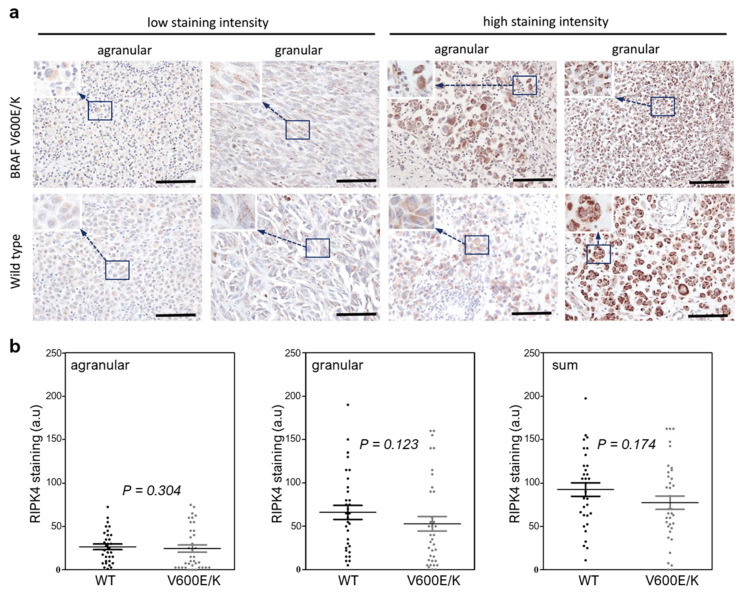
The BRAF mutation does not affect RIPK4 expression in metastatic melanoma. (**a**) Ex-pression of RIPK4 in wild-type and *BRAF*^V600E^ metastatic melanomas with low and high (agranular and granular stain pattern). Original magnification: 400x, scale bar = 100 µm. (**b**) Quantification of RIPK4 in melanoma specimens. The scatter plots present individual data for RIPK4 for each sample and the mean ± SD of wild-type (WT, n = 32) and (*BRAF* V600E/K, n = 36). Statistical analysis of 68 specimens using the Mann–Whitney rank sum test, as indicated.

**Figure 3 cancers-15-00918-f003:**
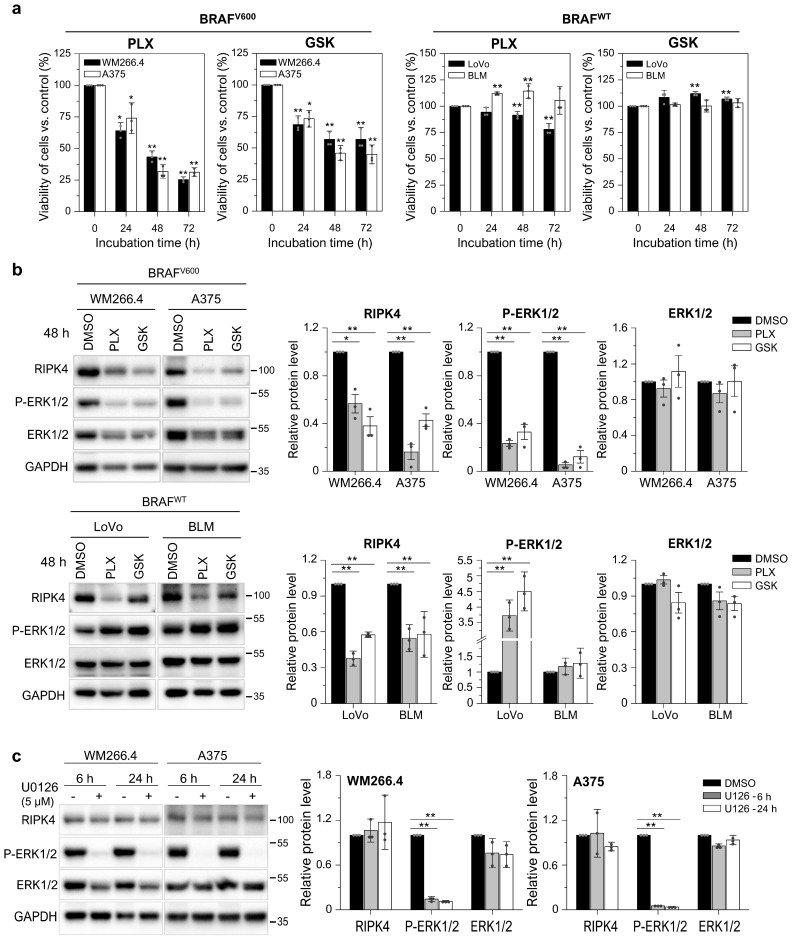
BRAF inhibitors decrease the level of the RIPK4 protein in BRAF^V600^ and BRAF^WT^ cells. (**a**) Viability of melanoma cells (A375, WM266.4, BLM) and LoVo cells treated with vemurafenib (PLX; 10 µM) and dabrafenib (GSK; 10 nM) at 24–72 h using the MTT assay. (**b**,**c**) Western blot images showing the levels of P-ERK1/2, ERK1/2 and RIPK4 proteins in cells after treatment with (**b**) vemurafenib (PLX; 10 µM) and dabrafenib (GSK; 10 nM) for 48 h or (**c**) U0126 (5 µM). GAPDH is the loading control. Each bar represents the mean ± SD of three biological replicates. Statistical analysis was performed using the Student’s t test. * *p* < 0.05 and ** *p* < 0.01 were considered significant.

**Figure 4 cancers-15-00918-f004:**
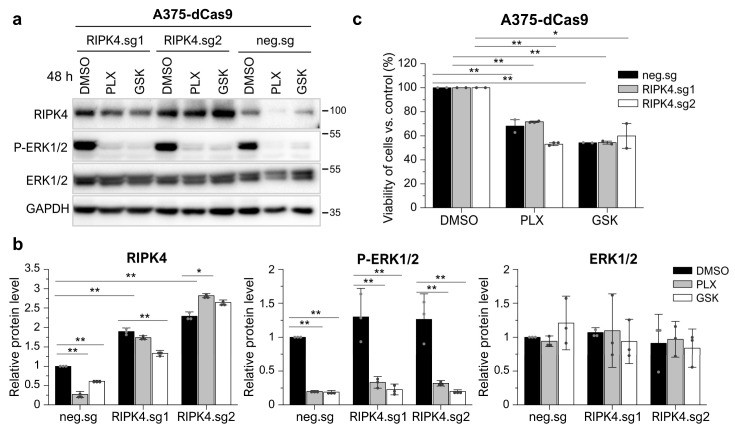
RIPK4 overexpression in A375-dCas9 cells inhibits the RIPK4 diminishing effect of BRAFi but do not affect ERK1/2 activity. A375-dCas9 cells are transfected with two single sgRNAs targeting RIPK4 (RIPK4.sg1, RIPK4.sg2) or no targeting control (neg.sg). After 72 h of transfection, cells were incubated with vemurafenib (PLX; 10 µM) and dabrafenib (GSK; 10 nM), or DMSO, as a control, for 48 h. (**a**,**b**) The expression levels of RIPK4, P-ERK1/2, and ERK1/2 were analyzed using Western blot along with densitometry. GAPDH was used as a loading control. (**c**) MTT assay after 24 h. Each bar represents the mean ± SD of three biological replicates. * *p* < 0.05 and ** *p* < 0.01 were considered significant.

**Figure 5 cancers-15-00918-f005:**
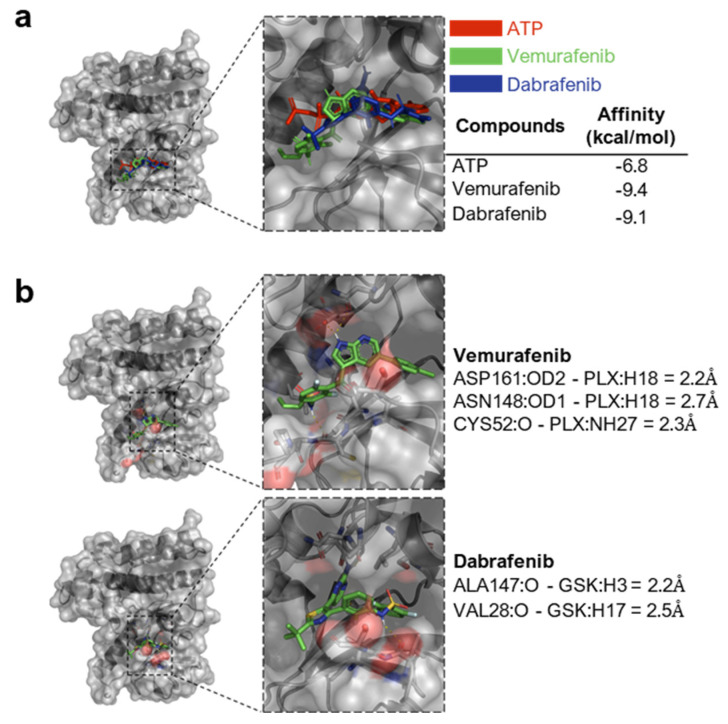
Inhibitors of the BRAF^V600^ protein can directly interact with the RIPK4 kinase. (**a**) Molecular docking of ATP, vemurafenib, and dabrafenib binding to RIPK4. RIPK4 was shown in cartoon style (with Connolly surface), while the analyzed ligand was shown in the stick model. The energy of affinity for vemurafenib, dabrafenib, and ATP to RIPK4 is −9.4; −9.1; −6.8 kcal/mol, respectively, for these inhibitors. (**b**) Potential hydrogen bonds between the RIPK4 ligand molecule and the protein.

**Figure 6 cancers-15-00918-f006:**
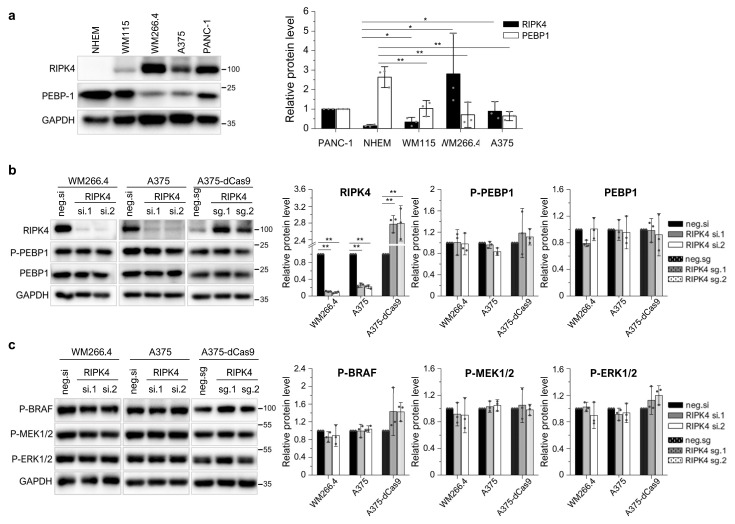
Downregulation and upregulation of RIPK4 do not affect PEBP1 or BRAF/ERK/MEK signaling pathway in melanoma cell lines. (**a**) Comparison of RIPK4 and PEBP-1 in melanocytes (NHEM), primary melanoma cells (WM115), and metastatic melanoma (WM266.4, A375) using Western blot along with densitometry, n = 3. PANC-1 cells expressed as a positive control. (**b**,**c**) Cells were transfected with RIPK4.si1 RNA, RIPK4.si2 RNA, or neg.si RNA and analyzed after 48 h via Western blot. Stable A375-dCas9-VPR cells were transfected with two single sgRNAs targeting RIPK4 (RIPK4.sg1, RIPK4.sg2) or with the negative control (sg neg) and analyzed after 72 h via Western blot. GAPDH is the loading control. (**b**) RIPK4, P-PEBP1, PEBP1 levels in WM266.4, A375, and A375-dCas9 cells. (**c**) P-BRAF, P-MEK1/2 and P-ERK1/2. Each bar represents the mean ± SD of 3 biological replicates. * *p* < 0.05 and ** *p* < 0.01 were considered significant.

**Figure 7 cancers-15-00918-f007:**
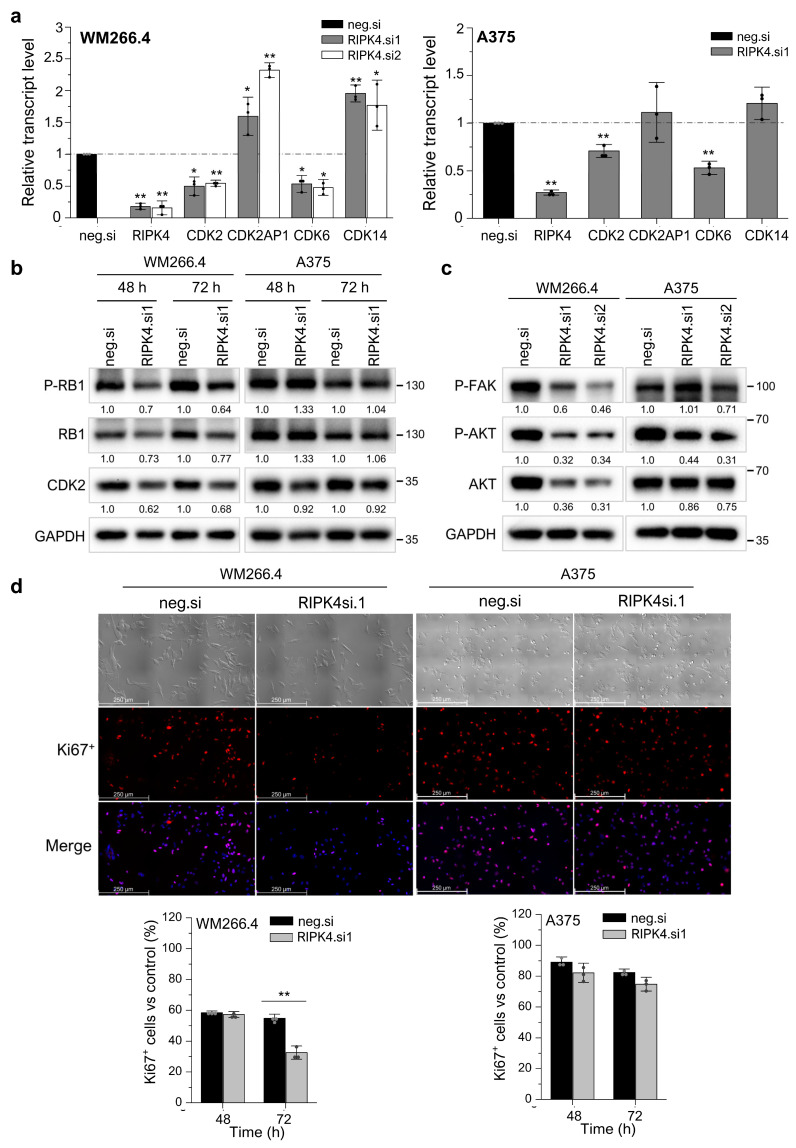
Downregulation of RIPK4 alters proliferation in WM266.4, but not in A375 cells. WM266.4 and A375 cells were transfected with RIPK4.si1, RIPK4.si2, or neg.si RNA and analyzed, n = 3. (**a**) Transcript levels of RIPK4, CDK2, CDK6, CDK14CDK2AP1 48 h after transfection normalized to GAPDH. (**b**,**c**) Western blot images showing the levels of (**b**) proteins P-RB1, RB1, and CDK2, (**c**) proteins P-FAK, p-AKT, and AKT proteins in A375 and WM266.4 cells. GAPDH is the loading control. (**d**) Phase-contrast images of cells and immunofluorescence images of Ki67 protein (red) and cell nuclei (blue) 72 h after transfection. Original magnification: 100×, scale bar = 250 µm. Each bar represents the mean ± SD of three biological replicates. All RIPK4.si transfected cells were compared with neg.si (scrambled control) samples. Statistical analysis was performed using the Student’s *t* test. * *p* < 0.05 and ** *p* < 0.001 were considered significant.

**Figure 8 cancers-15-00918-f008:**
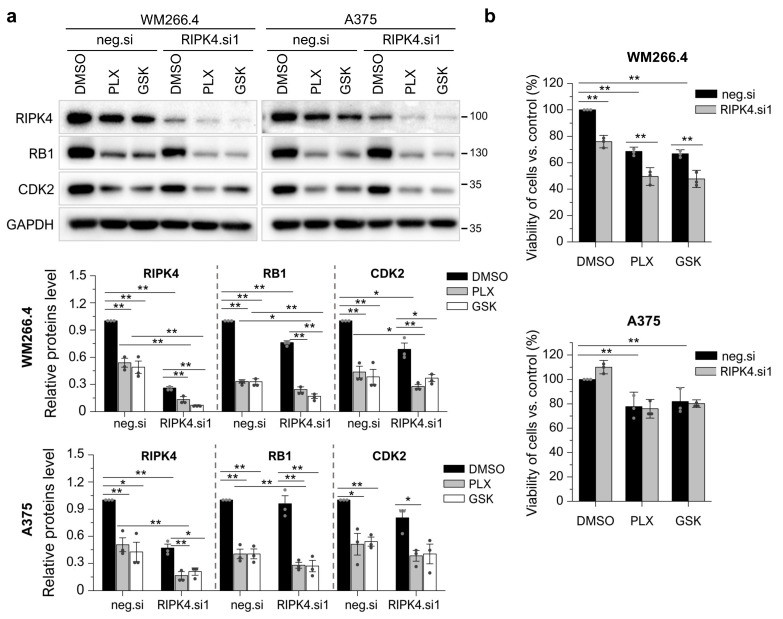
Downregulation of RIPK4 enhanced the effect of BRAFi on WM266.4 cell proliferation but does not A375. WM266.4 and A375 cells were transfected with RIPK4.si1 or neg.si RNA. After 48 h of transfection, cells were incubated with vemurafenib (PLX; 10 µM), dabrafenib (GSK; 10 nM), or DMSO, as a control, for 24 h. (**a**) The expression levels of the RIPK4, RB1, and CDK2 proteins were analyzed using Western blotting along with densitometry. GAPDH was used as a loading control. (**b**) Viability of the cells was assessed with MTT assay. Each bar represents the mean ± SD of three biological replicates. The samples were compared with the neg.si control treated with DMSO. * *p* < 0.05, ** *p* < 0.01.

**Figure 9 cancers-15-00918-f009:**
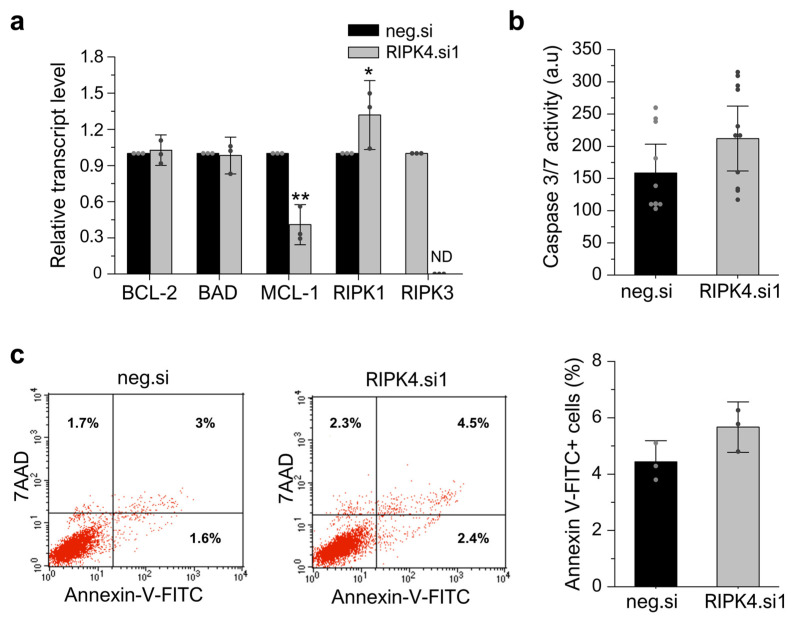
Silencing of RIPK4 using siRNA has no impact on apoptosis or necroptosis in WM266.4 cells measured 48 h after transfection. (**a**) Transcript levels of BCL-2, BAD, MCL-1, RIPK1, RIPK3 48 h after transfection with RIPK4.si1 or neg.si normalized to GAPDH. ND—not detected. (**b**) Caspase activity 3/7. (**c**) Representative Annexin V-FITC/7AAD double staining and FACS analysis (left). Annexin V-FITC positive cells (sum of Ann+/7AAD- and Ann+/7AAD+; right). Each bar represents the mean ± SD of three biological replicates. All RIPK4.si transfected cells were compared with neg.si (scrambled control) samples. Statistical analysis was performed using the Student’s *t*-test. * *p* < 0.05, ** *p* < 0.01 was considered significant.

**Figure 10 cancers-15-00918-f010:**
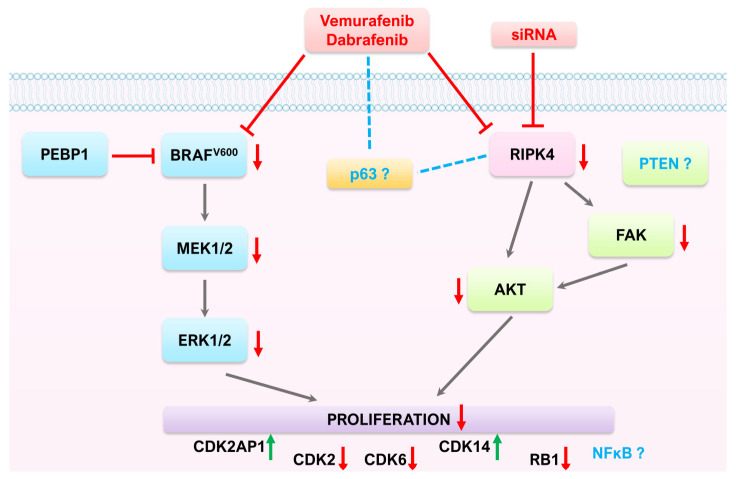
Proposed mechanisms of action of BRAFi (vemurafenib, dabrafenib) and RIPK4 on the RAF1/MEK/ERK pathway and proliferation in melanoma cells with BRAF^V600^.

**Table 1 cancers-15-00918-t001:** Patients’ characteristics.

Variable	Categorization	Analyzable n (%)
Age	<60 years	14 (20.6)
	>60 years	54 (79.4)
Gender	Female	42 (61.8)
	Male	36 (52.9)
Melanoma site	Primary melanoma	6 (8.8)
	Lymph node	41 (60.3)
	Recurrence	19 (27.9)
	Others	2 (2.9)
Treatment	Surgery	23 (35.3)
	Radiotherapy	4 (5.9)
	Chemotherapy/immunotherapy	30 (44.1)
	Radiotherapy + chemotherapy	10 (14.7)
	Surgery + radiotherapy	1 (1.5)
BRAF status	V600E	27 (39.7)
	V600K	5 (7.4)
	Wild-type	36 (52.9)
Mitotic index	≤5 mitoses/1 mm^2^	22 (32.3)
	>5 mitoses/1 mm^2^	46 (67.7)

## Data Availability

Data sharing not applicable. No new data were created or analyzed in this study. Data sharing is not applicable to this article.

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
