# Peer review of "Vemurafenib and Dabrafenib Downregulates RIPK4 Level"

_cancers, 2023, doi:10.3390/cancers15030918_

Round 1

Reviewer 1 Report

The paper titled “Vemurafenib and dabrafenib downregulates RIPK4 level” by Madej et al., is well written and the experiments are well designed. The data is easy to follow but can be improved.

I do not have any major concerns, but have some suggestions to improve the paper.

1.       The BRAF WT lines used are non-melanoma lines. BLM is a lung cancer line while LoVo is a colorectal cancer cell line. It will be nice to see a BRAF-WT melanoma line.

2.       For figure 9, did the authors look at the relative transcript level of MCL1? MCL1 is one of the major anti- apoptotic proteins in many solid cancers including melanoma and is more involved in the apoptosis process compared to BCL2.

Author Response

The paper titled “Vemurafenib and dabrafenib downregulates RIPK4 level” by Madej et al., is well written and the experiments are well designed. The data is easy to follow but can be improved.

I do not have any major concerns, but have some suggestions to improve the paper.

  1. The BRAF WT lines used are non-melanoma lines. BLM is a lung cancer line while LoVo is a colorectal cancer cell line. It will be nice to see a BRAF-WT melanoma line.

Thank you for this comment. We agree with the Reviewer that using melanoma cells without mutations in BRAF would be more interesting. However it should be noted that WT melanoma cells without mutations in BRAF are rare and finding one showing RIPK4 expression and no BRAF mutation is difficult.  Therefore, in addition to BLM, we used LoVo cells, in order to examine the effects of BRAF inhibitors on RIPK4 expression.

BLM are not lung cancer cells, they  have an amelanotic melanoma origin. BLM cells derived from melanoma metastasis to lung, according to the description of https://www.cellosaurus.org/CVCL_7035. 

  1. For figure 9, did the authors look at the relative transcript level of MCL1? MCL1 is one of the major anti- apoptotic proteins in many solid cancers including melanoma and is more involved in the apoptosis process compared to BCL2.

Thank you for this comment. As suggested by the Reviewer we tested the effect of RIPK4 downregulation on MCL1. It is significantly downregulated in WM266.4 cells. The data are included in Figure 9.

Reviewer 2 Report

The manuscript entitled “Vemurafenib and dabrafenib downregulates RIPK4 level” by Ewelina Madej et al.was presented as an original Article in which the authors analyze the effects of two different BRAF inhibitorsvemurafenib and dabrafenib, on RIPK4 in melanoma cells. The research work is comprehensive and systematic, and the text is well written, in a clear and organized manner. There are several major issues that need to be addressed before considering publication in Cancers:

Major comments for the text: 

A. The introduction needs to be revisedThe current format of the introduction is a little bit descriptiveMore novelty about this study should be described in the introduction part

B. In the Materials and Methods section, it is necessary indicate information about all reagents and equipment used (manufacturers/producers and catalog number)

C. The Results need to be revised because they must be mainly descriptive. More detailed background and references should be described in the introduction and/or discussion part (i.e. PAGE 13The CDK4/CDK6 pathway is frequently dysregulated in melanoma, leading to excessive proliferation. Cell cycle regulating kinases CDK4/CDK6 in complex with cyclin D1 and D2 or D3 can phosphorylate RB, thus causing the release of E2F transcription factors and the G1-S cell cycle transition [42]We recently found that silencing RIPK4 disrupts 398 the WM266.4 cell cycle [18]PAGE 14: Recently, it has been shown that CDK14 kinase expression may be under the control of the AKT pathway that controls cell proliferation, among many other things [44]. Increased signaling through the PI3K/AKT pathway, with or without concomitant MAPK reactivation, represents an alternative pathway for resistance to BRAFi treatment [14]).

D. Please, be consistent with naming dabrafenib (or GSK2118436) and vemurafenib (or PLX4032) throughout the text and the figure legends.

E. Please, double check and correct the typos of words/acronyms/abbreviations/tenses in the text. The reviewer suggests capitalizing the first letters of full words that define the acronyms/abbreviations.

F. Concerning the gene nomenclature, where the authors referred to gene, please replace it with own gene name.Moreover, gene name abbreviations must be italicized. 

Major comments for the data:

In general, the data presented in the figures are well organizedTherefore, although the conclusions are in line with the presented data, some results need to be improved including several experimental approaches and different methodologies. The authors evaluated the effects of RIPK4 inhibition on apoptosis/necroptosis in WM266.4 cells by measuring BCL-2 and BAD transcripts levels and Caspase 3/7 activity. AmRNA levels often do not reflect protein levels, these data could be improved analyzing the protein expression levels of anti- and pro-apoptotic markers (i.e.Bcl-2, Bax, or Caspases).

Please provide a Supplementary Figure in your Supplementary Information file to graphically account for the FACS sequential gating/sorting strategies (Annexin-V staining experiments) or provide gating/sorting strategies in-figure.

The original uncropped and unprocessed scans of all blots should be cited in the Materials and Methods section (2.8. Western Blotting)The reviewer suggests creating a table of antibodies used for immunofluorescence and Western Blot analyses, including the catalog number code, the concentration, and the hybridization conditions and a table of primers used for quantitative RT-PCR.

Major comments for the figures:

Figure 1c: please, specify the color code for BRAF (red), BRAFV600E (blue) and Ripk4 (yellow) also as legend in the figure.

Author Response

  1. The introduction needs to be revised. The current format of the introduction is a little bit descriptive. More novelty about this study should be described in the introduction part.

Thank you for this comment. The introduction has been revised and the novelty of the research has been highlighted. We wanted to underlined that data on RIPK4 in cancer are sparse and RIPK4 in melanoma was presented only in our previous paper (Madej at al, 2021). All chances are marked with red font.

2. In the Materials and Methods section, it is necessary indicate information about all reagents and equipment used (manufacturers/producers and catalog number).

Thank you for this comment. The catalog numbers have been added for all reagents. The names of the equipment manufacturer have been added.

3. The Results need to be revised because they must be mainly descriptive. More detailed background and references should be described in the introduction and/or discussion part

(i.e. PAGE 13: The CDK4/CDK6 pathway is frequently dysregulated in melanoma, leading to excessive proliferation. Cell cycle regulating kinases CDK4/CDK6 in complex with cyclin D1 and D2 or D3 can phosphorylate RB, thus causing the release of E2F transcription factors and the G1-S cell cycle transition [42]. We recently found that silencing RIPK4 disrupts 398 the WM266.4 cell cycle [18].

PAGE 14: Recently, it has been shown that CDK14 kinase expression may be under the control of the AKT pathway that controls cell proliferation, among many other things [44]. Increased signaling through the PI3K/AKT pathway, with or without concomitant MAPK reactivation, represents an alternative pathway for resistance to BRAFi treatment [14]).

Thank you for this comment. The text referring to published data has been removed.

4. Please, be consistent with naming dabrafenib (or GSK2118436) and vemurafenib (or PLX4032) throughout the text and the figure legends.

Thank you for this comment. In the manuscript, we used PLX and GSK abbreviations without numbers on the figures. The full names are too long and interfere with the figure. Thus in the legend we added an explanation of the abbreviation without the number. In the text we have corrected naming.

5. Please, double check and correct the typos of words/acronyms/abbreviations/tenses in the text. The reviewer suggests capitalizing the first letters of full words that define the acronyms/abbreviations.

Thank you for this comment. It has been checked and corrected. All corrections are marked with red font.

6. Concerning the gene nomenclature, where the authors referred to a gene, please replace it with own gene name. Moreover, gene name abbreviations must be italicized.

Thank you for this comment. It has been checked and corrected. All corrections are marked with red font.

Major comments for the data:

In general, the data presented in the figures are well organized. Therefore, although the conclusions are in line with the presented data, some results need to be improved including several experimental approaches and different methodologies.

Thank you for this comment. Since we observed no changes in mRNA level, instead of analyzing protein level we decided to test the activity of caspase 3/7 (evaluated at the protein level) and Annexin V /7AAD. Functional test provides more specific insights into the apoptosis process than change at the level of BAD, BCL2 proteins.

Please provide a Supplementary Figure in your Supplementary Information file to graphically account for the FACS sequential gating/sorting strategies (Annexin-V staining experiments) or provide gating/sorting strategies in-figure.

Thank you for this comment. We have added graphical for Annexin V/7AAD staining to Figure 9c, and the gating strategies  to raw data file.

The original uncropped and unprocessed scans of all blots should be cited in the Materials and Methods section (2.8. Western Blotting).

Thank you for this comment. We have added raw data from Western blotting to raw data file.

The reviewer suggests creating a table of antibodies used for immunofluorescence and Western Blot analyses, including the catalog number code, the concentration, and the hybridization conditions and a table of primers used for quantitative RT-PCR.

Thank you for this comment. Following the Reviewer's suggestion, we included the tables with antibodies and primers in the supplement: Table S1 and Table S2.

Major comments for the figures:

Figure 1c: please, specify the color code for BRAF (red), BRAFV600E (blue) and Ripk4 (yellow) also as legend in the figure.

Thank you for this comment. Figure 1c has been corrected.

Round 2

Reviewer 2 Report

The authors have addressed most of my concerns. The manuscript has been significantly improved and is now ready for the publication in Cancers in the current state.